# Pyrolysis of Mixed Plastic Waste: I. Kinetic Study

**DOI:** 10.3390/ma13214912

**Published:** 2020-10-31

**Authors:** Ibrahim Dubdub, Mohammed Al-Yaari

**Affiliations:** Chemical Engineering Department, King Faisal University, P.O. Box 380, Al-Ahsa 31982, Saudi Arabia; idubdub@kfu.edu.sa

**Keywords:** mixed plastic waste, recycling, pyrolysis, kinetics, thermogravimetric analysis (TGA), Coats-Redfern model, Criado model

## Abstract

Plastic wastes have become one of the biggest global environmental issues and thus recycling such massive quantities is targeted. Low-density polyethylene (LDPE), high-density polyethylene (HDPE), polypropylene (PP), and polystyrene (PS) are considered among the main types of plastic wastes. Since pyrolysis is one of the most promising recycling techniques, this work aims to build knowledge on the co-pyrolysis of mixed polymers using two model-fitting (Criado and Coats–Redfern) methods. Seventeen co-pyrolysis tests using a thermogravimetric analyzer (TGA) at 60 K/min for different mixed compositions of LDPE, HDPE, PP, and PS were conducted. It was observed that the pyrolysis of the pure polymer samples occurs at different temperature ranges in the following order: PS < PP < LDPE < HDPE. However, compared to pure polymer samples, the co-pyrolysis of all-polymer mixtures was delayed. In addition, the synergistic effect on the co-pyrolysis of polymer blends was reported. The Master plot of the Criado model was used to determine the most suitable reaction mechanism. Then, the Coats–Redfern model was used to efficiently obtain the kinetic parameters (R^2^ ≥ 97.83%) and the obtained values of the activation energy of different polymer blends were ranging from 104 to 260 kJ/mol. Furthermore, the most controlling reaction mechanisms were in the following orders: First order reaction (F1), Contracting sphere (R3), and then Contracting cylinder (R2).

## 1. Introduction

Globally, the current generation rate of municipal solid waste (MSW) has been reported as 2.01 Bt/year and it is estimated to reach 2.59 Bt/year by 2030 [1]. However, in Saudi Arabia, the reported generation rate of MSW in 2014 was is 15.3 Mt/year and it is expected to be double by 2033 [2]. Plastic wastes, as one of the major ingredients, represent 5.2% of the MSW in Saudi Arabia [3]. While incineration and landfill of waste plastics have some environmental issues, including harmful immissions and products [4], pyrolysis products are more promising with a low volume of produced gases [5]. However, to produce the desired type of oil, pyrolytic catalytic cracking is employed [6,7,8].

Extensive research on the pyrolysis of a single type of plastic waste such as high-density polyethylene (HDPE) [8,9], low-density polyethylene (LDPE) [9,10], polyvinyl chloride (PVC) [9,11,12,13], polypropylene (PP) [9,12,14], polycarbonate (PC) [15], and polystyrene (PS) [9,16,17] has been conducted. Although mixed plastic waste is the representative type of plastic waste worldwide, a limited number of works on the pyrolysis of mixed plastic waste plastic has been performed. In Saudi Arabia, plastic wastes contain mainly LDPE, HDPE, PS, PVC, and PP [18].

Although plastic recycling has attracted the attention of many researchers, it is economically and technically challenging to recycle mixed plastic wastes [19]. In addition, they cannot be easily separated from each other [20]. However, the reaction mechanism of the pyrolysis of mixed plastic may be different due to the interaction between the mixture constituents [19].

Chowlu et al. (2009) [17] studied the pyrolysis behavior of a mixture of two polymers PP and LDPE with five different mixture compositions and heating rates. The Vyazovkin model, a model-free technique, was used to investigate the effect of conversion on the thermal decomposition activation energy. The relation passed through three different zones: slow at low conversions, slightly high at the middle-range of conversions, and strongly high till the end of the decomposition. Low activation energy, which is preferred for any reaction, was reported for a mixture of 65 wt% PP/35 wt% LDPE.

Aboulkas et al. (2010) [21] studied the conversion model of the thermal decomposition of HDPE, LDPE, and PP by Coats–Redfern and Criado methods. It was reported that the Criado model described the “Contracting Sphere” model for HDPE and LDPE, and the “Contracting Cylinder” model for PP.

Silvarrey and Phan (2016) [4] developed a kinetic model to find the real reaction mechanism of the decomposition of five different polymers: HDPE, LDPE, PP, PS, and polyethylene terephthalate (PET) using TGA and MATLAB. Kissinger–Akahira–Sunose (KAS), Malek, and linear model fitting methods were used to predict the thermogravimetric analysis (TGA) data of the pyrolytic process of these five polymers. TGA data at four heating rates (5, 10, 20, and 40 K/min) covering the temperature range of 30–700 °C were reported. It was reported that all polymer wastes showed a similar thermogravimetric (TG) shape with one stage of thermal decomposition. However, the characteristics of TG temperatures (Tonset, Tpeak, and Tendset) were slightly different for the five tested plastic samples. The average values of the obtained activation energy were as follows: PS = 192.6 kJ/mol, PET = 197.6 kJ/mol, PP = 261.2 kJ/mol, LDPE = 267.6 kJ/mol, and HDPE = 202.4 kJ/mol.

Yu et al. (2016) [19] reviewed some of the published works investigating the pyrolysis of polyvinyl chloride (PVC) mixed with PP, polyethylene (PE), or PS. The effect of the added polymer on the mixture onset temperature, peak decomposition temperature, endset temperature, residue weight, and quantity, was reported. Different effects by different polymers were attributed to the nature of the added polymer.

Anene et al. (2018) [22] studied the thermal degradation of a mixture of LDPE and PP at different compositions. It was reported that pyrolysis started at lower temperatures for the LDPE/PP mixture when compared with pure LDPE, proving the interaction between the mixed polymers. However, the pyrolysis of pure PP started earlier than that of pure LDPE.

Mumbacha et al. (2019) [23] investigated the decomposition of plastic solid waste (PSW) by TG under inert conditions from 25 °C to 1000 °C with four heating rates (5, 10, 20, and 30 K/min). It was reported that the feedstock of PSW includes: 51.85 wt% PP, 17.28 wt% LDPE, 7.41 wt% HDPE, 17.28 wt% plastics with PVC, PET, and PS, and 6.18 wt% lignocellulosic. The kinetic parameters were obtained using four isoconversional methods (FWO, KAS, STK, and VYA models). However, the reaction model was identified by master plots. Three main reaction stages were observed. While the first decomposition reaction represented the main decomposition reaction of holocellulose and minor decomposition of the first degradation stage of PVC (dichlorination), the second decomposition reaction stands for the decomposition of a mixture of polymers, such as PS and some adhesive acrylic-based resins, and PVC (dichlorination), and the last decomposition reaction was mainly for the thermal decomposition of PP, LDPE and HDPE.

In addition, thermal degradation of PP with PE (LDPE and HDPE) blends were investigated to obtain the induction time [24]. However, the catalytic thermal degradation of the ternary mixture of PP/LDPE/HDPE was early studied by Himmelblau [25] in 1968. Furthermore, pyrolysis of PP with PE and PS was investigated [26] and the pyrolysis products were targeted.

As discussed earlier, there is a research gap on the pyrolysis of a representative mixed plastic waste needed to be further investigated. In addition, most of the available studies of the pyrolysis of the mixed polymers assume the first-order reaction mechanism which may not be the accurate model. Therefore, this work aims to build knowledge on the thermal decomposition of mixed polymers where two model-fitting techniques, namely Criado and Coats–Redfern Models, are used to obtain the best reaction mechanisms and kinetic parameters of the pyrolysis process using TGA data.

## 2. Materials and Methods

### 2.1. Materials

PP, PS, HDPE, and LDPE polymer samples were obtained from Ipoh SY Recycle Plastic Co., (Perak, Malaysia). The proximate analysis of all tested polymer samples was conducted using PerkinElmer Simultaneous Thermal Analyzer (STA) 6000, Waltham, MA, USA. The following steps were followed. The temperature was increased from the ambient temperature to 283 K at 10 K/min and under 20 mL/min N_2_ flowrate. Then, the temperature was held at 283 K for 10 min to determine the moisture content. Then, the temperature was increased by 10 K/min from 283 K to 1123 K, and then it was held at 1123 K for 10 min to determine the volatile content. After that, the gas was switched from N_2_ to O_2_ flows at 20 mL/min to determine the ash content. However, the ultimate analysis was performed using PerkinElmer 2400 Series II CHNS Elemental Analyzer, Waltham, MA, USA, and data of both tests are presented in Table 1.

### 2.2. Thermal Decomposition of Mixed Polymer Samples

Pellets of each polymer sample were ground by a grinding mill to produce polymer powder with an average particle size of 0.7 mm. Then, samples of powder mixtures were prepared as presented in Table 2.

Thermogravimetric analysis (TGA) was performed using the PerkinElmer thermogravimetric analyzer TGA-7(Perkin Elmer, Shelton, CT, USA) with a high precision weighing balance. For all TGA tests, experiments were conducted under N_2_ (99.999%) gas flowing at 100 mL/min. A constant heating rate of 60 K/min (moderate heating rate) was used to evaluate two model-fitting methods namely Coats–Redfern and Criado models. Powdered polymer samples of 11 mg mass were tested. Table 2 shows the experimental matrix of the compositions of the mixed polymer samples of 17 tests.

### 2.3. Kinetic Theory

Reaction rate (r) can be expressed as follows [10,14]:(1)$$r=\frac{d\alpha }{dt}=\beta \frac{d\alpha }{dt}=A_{o}exp(-\frac{E}{R\;T})f\left(\alpha \right)$$
where:*α*: is the reaction conversion,*t*: is the time (min),*β*: is the heating rate (K/min),*T*: is the temperature (K),*A_o_*: is the pre-exponential factor (min^−1^),*E*: is the activation energy (J/mol),*R*: is the universal gas constant (8.314 J/mol.K),*f*(*α*): is the concentration-dependent term
(2)$$\frac{d\alpha }{dT}=\frac{A_{o}}{\beta }exp(-\frac{E}{R\;T})f\left(\alpha \right)$$
Or
(3)$$\frac{d\alpha }{f\left(\alpha \right)}=\frac{A_{o}}{\beta }exp(-\frac{E}{R\;T})dT$$

Taking the integral of both sides:(4)$$g(\alpha )=\int_{0}^{a}\frac{d\alpha }{f\left(\alpha \right)}=\frac{A_{o}}{\beta }exp(-\frac{E}{RT})dT$$

Let $x=-\frac{E}{RT}$, $\frac{dx}{\mathrm{dT}}=\frac{E}{R}\;\frac{1}{T^{2}}$, and at *T* = 0: x = ∞. Then, Equation (4) can be written as follows:(5)$$g(\alpha )=\int_{0}^{a}\frac{d\alpha }{f\left(\alpha \right)}=\frac{A_{o}\;E}{\beta \;R}\int_{X}^{\infty }\frac{e^{-x}}{x^{2}}dx$$
(6)$$g(\alpha )=\frac{A_{o}\;E}{\beta \;R}p\left(x\right)$$

The *p*(*x*) is the temperature integral which is not easy to be obtained analytically. However, Equation (6) can be solved using either numerical integration or approximation. The difference between the various model-free methods is dependent on the type of approximation employed.

The Coats–Redfern method, which is an integral model-fitting method, applies an asymptotic series expansion for the temperature integral estimation. The final equation for this method [27] is:(7)$$ln\left[\frac{g\left(\alpha \right)}{T^{2}}\right]=ln\left[\frac{A_{o}\;R}{\beta \;E}\right]-\frac{E}{R\;T}$$

For the constant heating rate (*β*) and selected reaction mechanism (*g*(*α*)), plotting *ln*[*g*(*α*)/*T*^2^] against *1/T* will give straight-line correlation with slope and intercept of *−E/R*, and *ln*(*A_o_*
*R*/*β E*), respectively. The slope and the intercept can be used to calculate *E* and *A_o_*.

By combing Equation (1) with Equation (7), the Criado equation can be derived [21] and expressed as follows:(8)$$\frac{Z\left(\alpha \right)}{Z\left(0.5\right)}=\frac{f\left(\alpha \right)g(\alpha )}{f\left(0.5\right)g(0.5)}=\left(\frac{T_{\alpha }}{T_{0.5}}\right)\frac{{\left(\frac{d\alpha }{dt}\right)}_{\alpha }}{{\left(\frac{d\alpha }{dt}\right)}_{0.5}}$$
where:*f*(*α*): is the concentration-dependent term (shown in Table 3),*g*(*α*): is the concentration-dependent term (shown in Table 3),*T_α_*: Temperature at conversion α,*T*_0.5_: Temperature at conversion (α) = 0.5,(*dα/dt*)*_α_*: Conversion change with time at conversion (α), (*dα/dt*)_0.5_: Conversion change with time at conversion (α) = 0.5.

The left-hand side of Equation (8) (*f*(*α*) *g*(*α*)/*f*(0.5) *g*(0.5)) is called a reduced theoretical curve (*Z*(*α*)/*Z*(0.5)), which is the characteristic of each reaction mechanism, while the right-hand side can be obtained from the experimental data. An iterative comparison between these two sides will inform us which exact kinetic model will describe appropriately the reaction. Table 3 shows the common solid-state thermal reaction mechanisms, *f*(*α*) and *g*(*α*), used for the Coats–Redfern and Criado models [24,28]. Solid-state models are widely used to describe the reaction mechanism of solid starting materials (initially solid). Solids do not react at ambient conditions but rather they should be heated to higher temperatures and thus reaction will take place in the liquid phase. However, special care should be taken to find the appropriate model.

## 3. Results and Discussion

### 3.1. Thermal Analysis of Mixed Polymers

The thermogravimetric (TG) curves of pure polymer samples are shown in Figure 1. Generally, thermograms were similar and confirm a single degradation step for each polymer type with complete degradation and this is attributed to the negligible ash content as presented in Table 1.

As clearly shown in Figure 1 and Table 4, the TG characteristic temperatures (Tonset, Tpeak, and Tendset) of the pyrolysis of the tested polymer samples were in the following order: PS < PP < LDPE < HDPE. PP pyrolysis temperatures are expected to be lower than that of HDPE and LDPE due to the third carbon atom that decreases the polymer stability [28]. In addition, Hujuri et al. (2008) [29] reported that substituted and branched polymers like PS and PP decompose at lower temperatures when compared with linear polymers (LDPE and HDPE). Furthermore, LDPE has a higher degree of branching and thus lower density and thermal decomposition temperatures than HDPE. The temperature range of the pyrolysis processes was as follows: PS = 389–687 K, PP = 459–728 K, LDPE = 445–785 K, and HDPE = 444–800 K. The maximum decomposition rate was between 670–757 K.

For the co-pyrolysis of the plastic wastes, TG curves and the derivative thermogravimetric (DTG) analysis are shown in Figure 2, Figure 3 and Figure 4. Similarly, similar shapes of TG and DTG curves for the pyrolysis of different polymer mixtures were observed. Those curves also confirm a single step of the thermal degradation reaction for all tested mixture (binary, ternary, and quaternary) samples with complete degradation. However, as shown in Figure 2, the co-pyrolysis of polymer mixtures was shifted to higher thermal degradation temperatures than that of pure polymer samples. This effect is clear in the following order: PS > PP > LDPE > HDPE and this can be attributed to the possible interaction between the mixture constituents during decomposition [17].

In addition, as shown in Figure 3, the synergistic effect was observed for all binary polymer mixtures except LDPE/HDPE mixtures. Additionally, this finding can be attributed to the strength and the type of interaction between the two polymers during the degradation process [17].

Moreover, although the maximum degradation temperatures (T_Peak_), of all the PS/HDPE, PS/LDPE, PP/HDPE were lying between the T_peak_ of both pure polymer samples, the PS/PP samples at different compositions have higher T_peak_ than those of pure PS and pure PP as shown in Figure 4. Similarly, the PP/LDPE sample with a mass ratio of 30/70 has a T_peak_ higher than the ones for pure polymers. This is another proof of the observed synergistic effect which can be due to the transfer of a hydrogen atom from the less stable polymer to the other during the pyrolysis process [30]. Alternatively, the synergistic effect can be investigated by comparing the experimental results with estimated ones based on the additive rule. Different behaviors of polymers in the mixtures were reported to lead to complex decomposition and the interaction between different polymers depends mainly on the miscibility of each polymer in the mixture [17]. Further investigation is highly recommended to explore the synergistic phenomenon at low heating rates.

The thermograms maximum temperatures (T_peak_), obtained from the derivative thermogravimetric (DTG) curves presented in Figure 5, are tabulated in Table 5.

For ternary polymer mixtures (Figure 5e), it is interesting to mention that all DTG curves were shifted to a higher temperature in the following order (test 13 ˂ test 15 ˂ test 14 ˂ test 16) which can be attributed to the increase in the content of PE polymers (LDPE and HDPE) where the PE content was 33.3%, 34.9%, 51.7%, and 70.5% for the tests 13, 15, 14, and 16, respectively. Additionally, Figure 5f confirms the same observation where the DTG curves were shifted to a higher temperature in the following order (test 13 ˂ test 17) when the total composition of both LDPE and HDPE was increased from 33.3 wt% to 50 wt%. PE polymers are more stable compared with PP, and PS, and thus degrade at higher temperatures.

### 3.2. Determination of Reaction Mechanisms and Kinetic Parameters

In this work, two model-fitting models—Criado and Coats–Redfern—were used to obtain reaction mechanisms, and kinetic parameters, respectively. The Criado equation is called sometimes the master plot since it is used to determine the most appropriate reaction mechanism by using master plots as shown in Figure 6 and Figure 7. Based on these figures, the most appropriate kinetic mechanism for each test was determined. Plots of kinetic mechanisms D1, R1, P2, P3, and P4 were not presented in Figure 6; Figure 7 due to the weak performance of those models compared with the other models.

After the determination of the best reaction mechanism (*g*(*α*)) for each test, the Coats–Redfern model was used to obtain the kinetic parameters (*E* and *A_o_*) by plotting *ln*[*g*(*α*)/*T^2^*] against 1/T (Figure 8). Since the obtained *E* values by the A2, A3, and A4 reaction mechanisms were very far from the expected and published ones, these values were ignored. Table 6 shows the values of *E*, *ln* (*A_o_*), and *R^2^* for each test. Values of *E* are ranging from 104 to 289 kJ/mol which depends on the selected model of reaction mechanism. In addition, as presented in Table 6, the most controlling reaction mechanisms are in the following order: First order reaction (F1), Contracting sphere (R3), and then Contracting cylinder (R2).

## 4. Conclusions

The TG and DTG thermograms obtained from the TGA study showed similar shapes and trends for all pure (LDPE, HDPE, PP, and PS) and mixed polymer samples with different compositions. From the TGA data, it was observed that the data conform only to a single thermal degradation step and pyrolysis of the pure polymer samples occurs at different temperature ranges in the following order: PS < PP < LDPE < HDPE. However, the co-pyrolysis of all-polymer mixtures was delayed when it was compared with the pyrolysis of pure polymer samples. In addition, the synergistic effect of the co-pyrolysis of some polymer blends was observed. However, further investigation is highly recommended to explore more the synergistic phenomenon at a low heating rate.

Furthermore, two different model-fitting methods were used to determine the most suitable reaction mechanism for each test and then to obtain the kinetic parameters for each reaction. The obtained values of activation energy were ranging from 104 to 260 kJ/mol depending on the properly selected model reaction mechanism. The most controlling reaction mechanisms were in the following order: First-order reaction (F1), Contracting sphere (R3), and then Contracting cylinder (R2).

## Figures and Tables

**Figure 1 materials-13-04912-f001:**
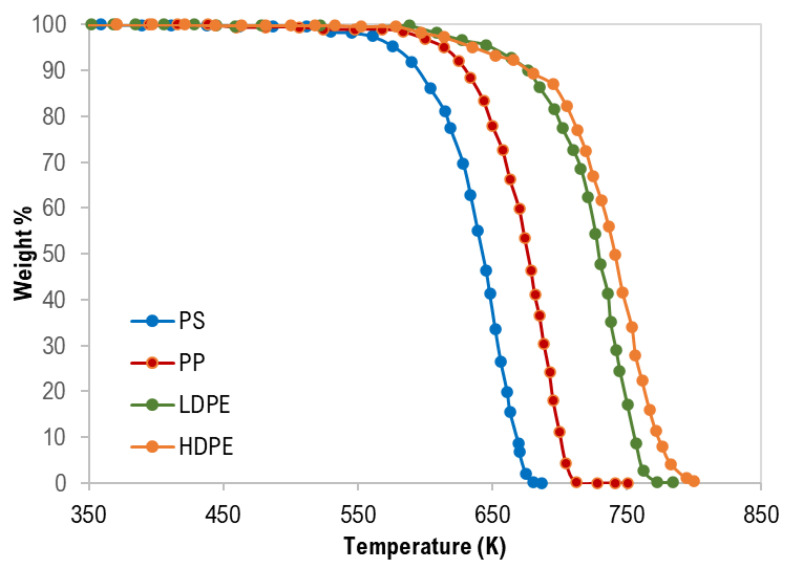
Thermogravimetric (TG) curves of pure polymer samples.

**Figure 2 materials-13-04912-f002:**
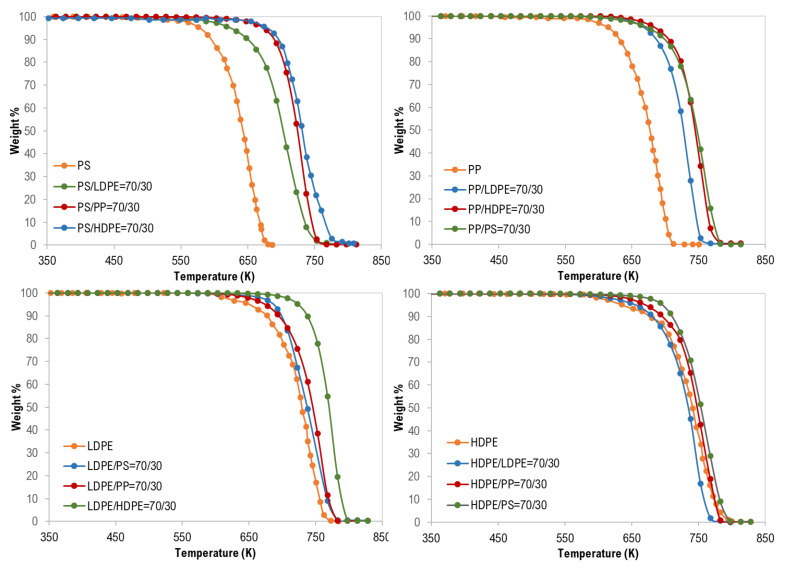
Thermogravimetric (TG) curves of binary mixture polymer waste.

**Figure 3 materials-13-04912-f003:**
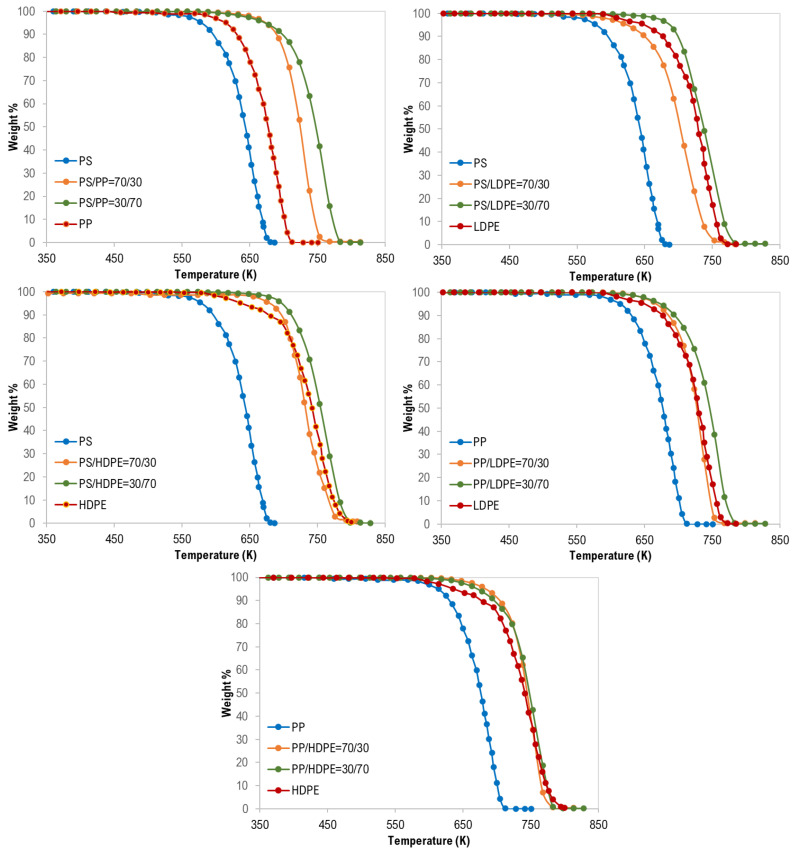
Synergistic effect on the co-pyrolysis of polymer samples.

**Figure 4 materials-13-04912-f004:**
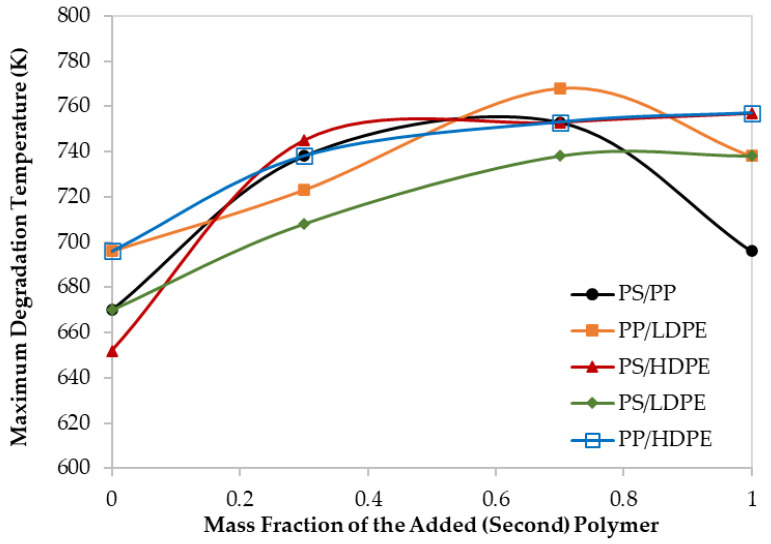
Maximum decomposition temperature for different polymer blends at 60 K/min.

**Figure 5 materials-13-04912-f005:**
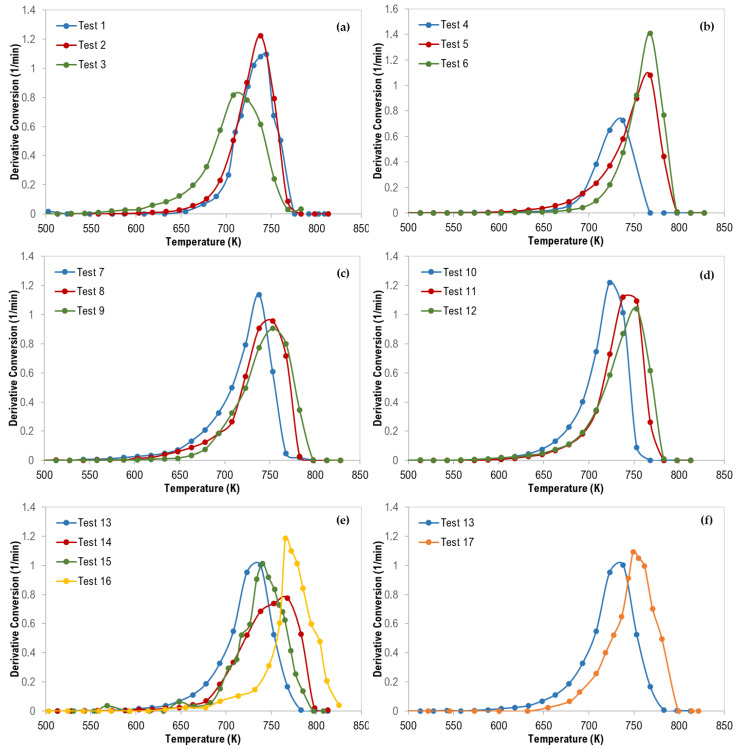
Derivative Thermogravimetric (DTG) curves of polymer mixtures: (**a**) 70 wt% PS, (**b**) 70 wt% LDPE, (**c**) 70 wt% HDPE, (**d**) 70 wt% PP, (**e**) ternary mixtures, and (**f**) ternary (with 33% each—Test 13) and quaternary (with 25% each—Test 17) mixtures.

**Figure 6 materials-13-04912-f006:**
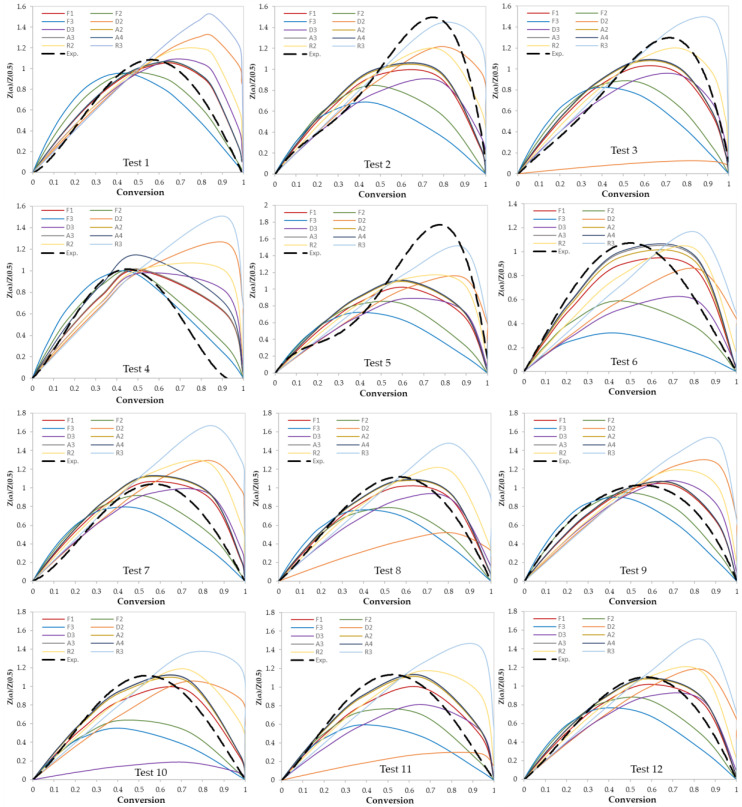
Criado model master plots of binary polymer mixtures.

**Figure 7 materials-13-04912-f007:**
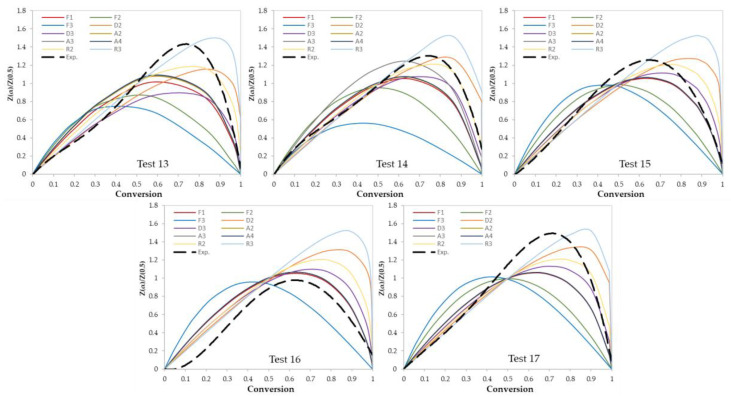
Criado model master plots of ternary and quaternary polymer mixtures.

**Figure 8 materials-13-04912-f008:**
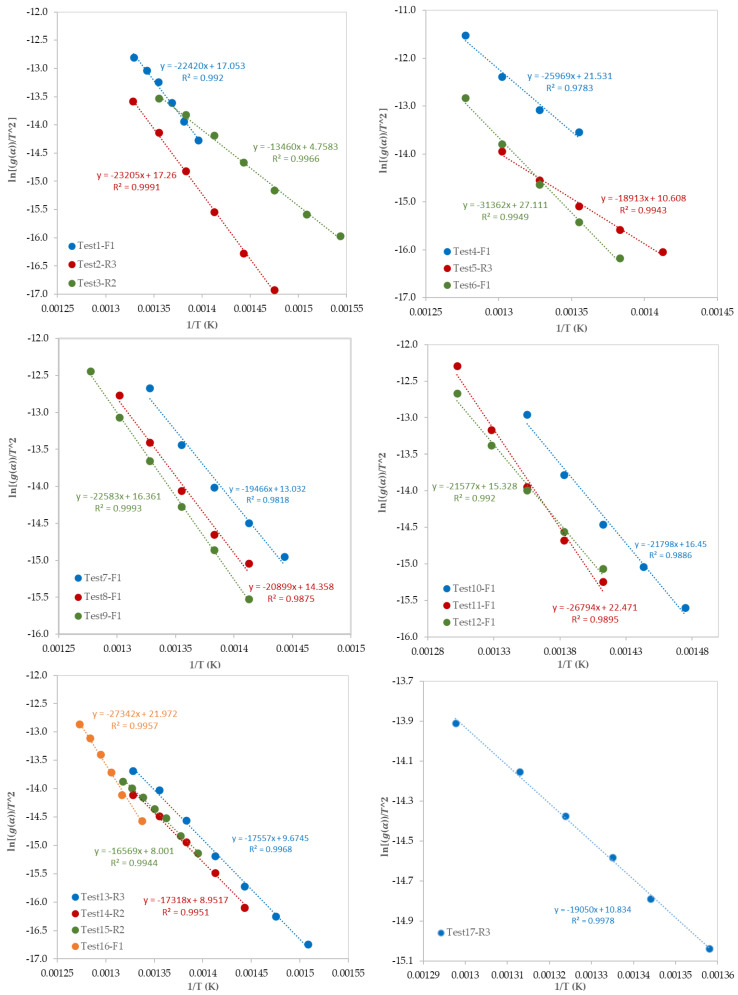
Plots of the Coats–Redfern model.

**Table 1 materials-13-04912-t001:** Ultimate and proximate analysis of different waste plastics.

Plastic	Proximate Analysis, wt%			Ultimate Analysis, wt%			
	Moisture	Volatile	Ash	C	H	N	S
**PP**	0.076	99.630	0.294	85.00	14.73	0.04	0.23
**PS**	0.235	99.590	0.175	90.47	9.43	0.00	0.08
**HDPE**	0.405	99.377	0.218	82.77	16.92	0.00	0.29
**LDPE**	0.199	99.653	0.148	83.00	16.75	0.00	0.25

**Table 2 materials-13-04912-t002:** Polymers compositions of the experimental tests.

Test No.	Mass Fraction, %				Test No.	Mass Fraction, %			
	PS	LDPE	HDPE	PP		PS	LDPE	HDPE	PP
1	70	0	30	0	10	0	30	0	70
2	70	0	0	30	11	0	0	30	70
3	70	30	0	0	12	30	0	0	70
4	30	70	0	0	13	33.3	0	33.3	33.3
5	0	70	0	30	14	48.4	19.2	32.5	0
6	0	70	30	0	15	48.4	34.9	0	16.7
7	0	30	70	0	16	0	15.5	55.0	29.5
8	0	0	70	30	17	25.0	25.0	25.0	25.0
9	30	0	70	0					

**Table 3 materials-13-04912-t003:** Common solid-state thermal reaction mechanisms.

Reaction Mechanism	*f*(*α*)	*g*(*α*)
First-order reaction (F1)	1−α	−ln (1−α)
Second order reaction (F2)	(1−α)^2^	[1/(1−α)] −1
Third order reaction (F3)	(1−α)^3^	{[1/(1−α)^2^] −1}/2
One dimensional diffusion (D1)	1/(2 α)	α^2^
Two dimensional diffusion (D2)	1/[−ln (1−α)]	(1−α) ln(1−α) + α
Three dimensional diffusion (D3)	3/{2[1−(1−α)^1/3^]}	[1−(1−α)^1/3^]^2^
Avrami–Erofeev (A2)	2(1−α)[−ln(1−α)]^1/2^	[−ln(1−α)]^1/2^
Avrami–Erofeev (A3)	3(1−α)[−ln(1−α)]^2/3^	[−ln(1−α)]^1/3^
Avrami–Erofeev (A4)	4(1−α)[−ln(1−α)]^3/4^	[−ln(1−α)]^1/4^
Phase boundary—one dimension (R1)	1	α
Contracting cylinder (R2)	2(1−α)^1/2^	1−(1−α)^1/2^
Contracting sphere (R3)	3(1−α)^1/3^	1−(1−α)^1/3^
Power low (P2)	2 α^1/2^	α^1/2^
Power low (P3)	3 α^2/3^	α^1/3^
Power low (P4)	4 α^3/4^	α^1/4^

**Table 4 materials-13-04912-t004:** Thermogravimetric analysis data of pure polymer samples at 60 K/min.

Polymer Sample	T_onset_ (K)	T_peak_ (K)	T_endset_ (K)	∆T (K)
PS	545	652	680	135
PP	583	696	710	127
LDPE	608	738	770	162
HDPE	614	757	790	176

**Table 5 materials-13-04912-t005:** Maximum decomposition temperature for different polymer blends at 60 K/min.

Test No.	T_peak_ (K)	Composition (wt %)
1	745	PS/HDPE (70/30)
2	738	PS/PP (70/30)
3	708	PS/LDPE (70/30)
4	738	LDPE/PS (70/30)
5	768	LDPE/PP (70/30)
6	768	LDPE/HDPE (70/30)
7	748	HDPE/LDPE (70/30)
8	753	HDPE/PP (70/30)
9	753	HDPE/PS (70/30)
10	723	PP/LDPE (70/30)
11	738	PP/HDPE (70/30)
12	753	PP/PS (70/30)
13	738	PS/HDPE/PP (33.3/33.3/33.3)
14	768	PS/LDPE/HDPE (48.4/19.2/32.5)
15	741	PS/LDPE/PP (48.4/34.9/16.7)
16	766	LDPE/HDPE/PP (15.5/55/29.5)
17	749	PS/PP/LDPE/HDPE (25/25/25/25)

**Table 6 materials-13-04912-t006:** Kinetic parameters of the pyrolysis of mixed polymers obtained by the Coats–Redfern model.

Test No.	Kinetic Parameters			Reaction Mechanism
	*E* (kJ/mol)	*ln* (*A_o_*)	R^2^	
1	186	31.16	0.992	First order reaction (F1)
2	193	31.41	0.9991	Contracting sphere (R3)
3	104	16.92	0.9966	Contracting cylinder (R2)
4	216	35.79	0.9783	First order reaction (F1)
5	157	24.55	0.9943	Contracting sphere (R3)
6	260	41.56	0.9949	First order reaction (F1)
7	158	26.98	0.9818	First order reaction (F1)
8	174	28.4	0.9875	First order reaction (F1)
9	188	30.48	0.9993	First order reaction (F1)
10	181	30.53	0.9886	First order reaction (F1)
11	223	36.76	0.9895	First order reaction (F1)
12	179	29.4	0.992	First order reaction (F1)
13	146	23.54	0.9968	Contracting sphere (R3)
14	144	22.81	0.9951	Contracting cylinder (R2)
15	138	21.81	0.9944	Contracting cylinder (R2)
16	227	36.28	0.9957	First order reaction (F1)
17	158	24.78	0.9978	Contracting sphere (R3)

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
