# Peer review of "Pyrolysis of Mixed Plastic Waste: I. Kinetic Study"

_materials, 2020, doi:10.3390/ma13214912_

Round 1
Reviewer 1 Report
- Line 31, Page 1: Data given in this statement is misleading. The reference cited for this data has no reference in their text. Please check.
- Line 40, Page 2: Statement needs to be reframed and reference cited contains no such data. Every region on the globe have different MPW stream content. Therefore, it is highly desired to give geographical address in the statement. Please check.
- Line 59-60, Page 2: Please reframe the sentence.
- Line 108, Page 3: Conditions for thermogravimetric analysis is crucial for the present study. Therefore, it would be suitable to mention all the conditions in experimental section instead of refereeing to other cited articles.
- It is not clear from the experimental section that samples were the blend or just mixture of pellets. If blend, how it was prepared? Please give information.
Author Response
Dear Respected Reviewer,
Thanks for your valuable comments that lead to a significant improvement of the manuscript.
Please find attached the file of our response to your comments.
Thanks again with best regards

Reviewer 2 Report
I think that sophisticated thermal methods will be the future solution for recycling mixed polymer wastes. Therefore I find the topic interesting and necessary.
A few notes and suggestions for the authors:
Line 10: "annoying" - not the best english term, replace
L13: "Since pyrolysis is one of the most preferable.." - most preferable by whom?
L23-L26: This is probably not necessary in the abstract as the Criado model has not been opened yet .
L33: "..waste plastics have some environmental issues" - I find this an understatement and not well in line with the original, quite old source ("Although combustion is an alternative to disposal in rubbish dumps (where harmful methane is produced),it must be subjected to severe environmental controls in order to meet the legal restrictions concerning the emission of solid particles and gaseous effluent")
L100: "Recycled PP, PS, HDPE, and LDPE polymer samples were obtained.." - Were these polymers chosen based on their prevalence or why?
L169: "..samples shifted 169 to higher temperatures (Tonset, Tpeak, and Tendset) in the following order: PS<PP<LDPE<HDPE." - This is a bit unclear expression - from what point to where does the pyrolysis temperature of pure polymers shift to? Compared to what?
L182: "the co-pyrolysis of polymer mixtures was shifted to higher thermal degradation temperatures than that of pure polymer samples." - this statement makes perfect sense.
L206: "Further investigation is highly recommended to explore the synergistic phenomenon." - I fully agree, this is important for future recycling of mixed polymeric wastes.
L247 & L249 - the figures: Is the conversion % really from 0% to 1% or should it be from 0 to 100%?
L252: "..study showed the same shapes and trends.." - I would not use this phrase, as the graphs are not the same - they may have similar shapes but are not the same, as you mention later.
Author Response

(The authors gave the same response as above.)

Reviewer 3 Report
There is great concern on the correctness of presented TG results since the reported temperatures and strong stabilization in mixtures are not realistic.
Why the study was performed on recycled polymers instead of fresh ones? Impurities and structural/compositional defects in recycled materials have strong effects on thermal degradation. For example, HDPE shows decreased stability in the 580 – 680 K range compared with LDPE. Also, both PE samples show lower Tonset compared with PP. Therefore observed behavior is due not only to polymers themselves but also to contaminants/defects in recycled materials.
Particular comments:
2.1: Experimental conditions and the way of performing proximate analysis through STA should be presented. Ash normally refers to inorganic content of a material, while the solid residue remaining at high temperature after TG measurements might contain undecomposed carbonaceous residue. Shift from inert gas to oxygen after reaching the plateau at the end of TG is necessary to burn potentially remaining carbon, so that remaining material is inorganic ash. Are 3 digits relevant in Table 1? Closed balance (100 %) should be checked for proximate analysis. Humidity of 4.5 % for HDPE is not observed in TG curve from Figure 1.
2.2: The nature (N2?) and flow of the gas used in TG analysis should be mentioned. IT should be mentioned how were polymers transformed into powder, what is the average particle size and how were prepared the polymer mixtures before TG analysis to ensure proper mixing of components for low amounts of 11 mg sample. What is the reason for using 60 K/min as heating rate? Fast heating is not favourable for the observation of interactions in mixtures.
Table 2.2: The selection of polymer ratio in mixtures should be explained. Interactions are difficult to be observed in unbalanced compositions such as 7/3 in binary mixtures.
Table 3: Relevance of using solid-state reaction mechanism to describe pyrolysis processes occurring in liquid (melted) phase should be described.
Line 167: One cannot consider “identical” the thermograms in Figure 1. “Single degradation step” instead on “single degradation reaction”.
Figure 1 and Table 4: Is it about “pure” polymers or “individual” recycled polymers? Temperature range should be extended to include at least the onset of the processes (as low as 388 K in Table 4). Table 4: 0% residue contradicts ash amount in Table 1. Temperature should be checked since according to Table 4 thermal degradation occurs over a temperature range (Tendset-Tonset) of 250 – 350 K, which is not reasonable, especially at fast heating rates. Also, reported Tonset are between 389 and 459 K, which means 115-185 oC, which is totally out of reality for PS, PP, PE. Strong stabilization of both components in binary mixtures (e.g. shift with ~ 50K to higher temperatures for LDPE/HDPE 70/30 in Figure 3) is totally unrealistic.
Figure 5: Run numbers is meaningless. Synergistic effects should be discussed by comparing the experimental results with estimated ones based on additive rule.
Table 6: CR plots should be given, to confirm linearity, despite R2 values.
Author Response

(The authors gave the same response as above.)

Round 2
Reviewer 1 Report
Now, this paper can be published.
Author Response
Many thanks with best regards

Reviewer 3 Report
There is a serious concern that changes in the manuscript are only rewritings to fit reviewer’s comments, while the fundamental aspects were in fact not touched. There is no real improvement of the manuscript.
Q1: Removing the world “recycling” does not change the basics; once they are recycled materials, supplemental aspects appear, that are still not discussed.
Q2.1: It appears now, from lines 104-108, that proximate analysis was performed at 10 K/min with gas switch from N2 to O2, while kinetic studies were performed at 60 K/min (which is a fast heating rate in TG, not a moderate one). TG and DTG curves should be provided for individual polymers for both heating rates, differences observed from the two rates should be discussed, explanation should be given, based on literature data, on why 60 K/min was used instead of 10 K/min for kinetic studies.
Table 2.2. Compositions of 70/30 are more closer to 3/1 (75/25) than to 1/1 (50/50).
Figures 1-3: The extension of x axis to lower temperatures of 350 K is meaningless now since much higher Tonset (above 610 K) were reported. The meaning was to include the T onset.
Table 4: Tonset is again improperly determined. One cannot call “onset” a temperature at which mass loss is of ~ 15 %, as observed at all newly reported Tonset values.
Figure 5: Replacing “Run” with “Test” does not help in easy distinguish the mixture composition.
Table 5: Additive rule applies to whole TG / DTG curves, not to Tpeak. Why Tests 6 and 7 are not considered synergistic?
Figure 8: Visual inspection of CR plots show that they are not linear, despite high R2 values. Linearity properly applies when points are randomly placed around the median value. However, most curve in Figure 8 are either concave/convex (ex. Tests 4, 7, 10 to mention the clearest ones) or sigmoid (ex. Tests 3, 13). This is a proof that simple mathematical description is not enough.
Author Response
Dear Respected Reviewer,
Greetings
Please find attached the requested clarifications.
Thanks with best regards
